# Construction of IL-13 Receptor α2-Targeting Resveratrol Nanoparticles against Glioblastoma Cells: Therapeutic Efficacy and Molecular Effects

**DOI:** 10.3390/ijms221910622

**Published:** 2021-09-30

**Authors:** Xiao-Min Lin, Xiao-Xiao Shi, Le Xiong, Jun-Hua Nie, Hai-Shan Ye, Jin-Zi Du, Jia Liu

**Affiliations:** 1Research Center, South China University of Technology (SCUT) School of Medicine, Guangzhou 510006, China; mcsmall@mail.scut.edu.cn (X.-M.L.); 201710106932@mail.scut.edu.cn (L.X.); mcnie@mail.scut.edu.cn (J.-H.N.); 201920153831@mail.scut.edu.cn (H.-S.Y.); 2Institute of Life Sciences, South China University of Technology (SCUT) School of Medicine, Guangzhou 510006, China; 201720151205@mail.scut.edu.edu.cn (X.-X.S.); jzdu@scut.edu.cn (J.-Z.D.); 3Liaoning Laboratory of Cancer Genetics and Epigenetics, Dalian Medical University, Dalian 610044, China

**Keywords:** glioblastoma, resveratrol, IL-13Rα2, Pep-PP@Res, JNK signaling

## Abstract

Glioblastoma multiforme (GBM) is the most common lethal primary brain malignancy without reliable therapeutic drugs. IL-13Rα2 is frequently expressed in GBMs as a molecular marker. Resveratrol (Res) effectively inhibits GBM cell growth but has not been applied in vivo because of its low brain bioavailability when administered systemically. A sustained-release and GBM-targeting resveratrol form may overcome this therapeutic dilemma. To achieve this goal, encapsulated Res 30 ± 4.8 nm IL-13Rα2-targeting nanoparticles (Pep-PP@Res) were constructed. Ultraviolet spectrophotometry revealed prolonged Res release (about 25%) from Pep-PP@Res in 48 h and fluorescent confocal microscopy showed the prolonged intracellular Res retention time of Pep-PP@Res (>24 h) in comparison with that of free Res (<4 h) and PP@Res (<4 h). MTT and EdU cell proliferation assays showed stronger suppressive effects of Pep-PP@Res on rat C6 GBM cells than that of PP@Res (*p* = 0.024) and Res (*p* = 0.009) when used twice for 4 h/day. Pep-PP@Res had little toxic effect on normal rat brain cells. The in vivo anti-glioblastoma effects of Res can be distinctly improved in the form of Pep-PP@Res nanoparticles via activating JNK signaling, upregulating proapoptosis gene expression and, finally, resulting in extensive apoptosis. Pep-PP@Res with sustained release and GBM-targeting properties would be suitable for in vivo management of GBMs.

## 1. Introduction

Glioblastoma multiforme (GBM) is the most common primary brain malignancy with extremely poor prognosis [1,2]. Surgical resection combined with radiotherapy and/or chemotherapy is the preferred treatment for GBMs [3]. Nevertheless, the current median survival time of GBM patients is only 12–16 months, and the five-year survival rate is less than 10% [4,5]. Due to the functional and anatomic particularities of the brain, it is extremely difficult to radically remove the aggressive tumor with minimal brain damage [6]. Combination of tumor treatment fields (TTField) with temozolomide (TMZ) increases two-year relative survival (RS) rates of GBM patients [7], while persistent administration of TMZ may cause secondary drug resistance, resulting in tumor recurrence and patient death [8]. It is thus urgently needed to find new drugs with low toxicity and better therapeutic effects against GBMs.

It has been well-documented that resveratrol (Res) as a natural polyphenolic compound has many beneficial biological activities, including anticancer and chemoprophylactic effects [9,10]. It inhibits growth and promotes differentiation and apoptosis of cancer cells including GBM cells through causing multiple molecular events, of which suppressed STAT3 activation and increased ROS generation are known to be critical [9,11]. More importantly, Res can penetrate the blood–brain barrier by simple diffusion, and the anti-GBM concentrations of Res have little toxic effect on normal neurons and glial cells because of the intact intracellular metabolic machinery for resveratrol in normal cells [11,12]. On the other hand, this well-operated system allows normal cells to easily biotransform Res and then quickly eliminate the metabolized Res products from them and finally from the body. Consequently, it is difficult for the systemically administered Res to reach effective concentrations in the tumor region [13]. In this context, it is necessary to design a resveratrol formulation with sustained release and GBM-targeting capacities for the practical use of resveratrol against GBMs.

Nanoparticles (NPs) have been employed for the selective delivery of drugs to the tumor site because of their ability to accumulate in the solid tumor mass through passive (enhanced permeability and retention effects; EPR) and/or active (linking to specific tumor markers and identifiers) targeting [14,15]. So far, a number of drug carriers such as chitosan nanoparticles [16], liposomes [9], solid lipid nanoparticles (SLN) [17] and lipid core nanocapsules (LCN) [18] have been developed and used to package Res. Nanoparticle formulations, especially transferrin-modified nanoparticles, exert a more significant anti-GBM effect than free Res and result in improved survival of C6 tumor xenograft-bearing rats [9]. It thus indicates that a combination of passive and active targeting may greatly improve therapeutic efficacy. To reach this goal, adding a reliable GBM biomarker to Res-harboring NPs would be required. It has been reported that IL-13Rα2, a 65 kD plasma membrane receptor, is overexpressed in GBM tissues [19] and mediates endocytosis after binding to its ligand [20]. Moreover, IL-13Rα2 is regarded as a potential target in GBM-oriented treatment because it is absent or expressed at an extremely low level in normal brain tissue [21]. Pep-1, a short peptide comprising nine amino acids (CGEMGWVRC), has been confirmed to bind and home to both subcutaneous and orthotopic GBM xenografts expressing IL-13Rα2 [22,23]. In this study, we synthesized Res-loaded poly(ethylene glycol)-*b*-poly(caprolactone) (PEG-*b*-PCL; PP) and modified it with Pep-1 so as to solve the administration shortcomings of Res for better treatment of GBMs and in-depth understanding of the molecular events caused by the new Res formulation.

## 2. Results

### 2.1. IL-13Rα2 Expression in C6 Cells

IL-13Rα2-oriented immunocytochemical staining was performed on C6 cells as well as a case of human GBM. The normal rat brain tissue and the astrocytes isolated from 2D+ cultured rat brain tissue were used as normal controls. IL-13Rα2-positive labeling was observed in the cytoplasm and on the membranes of both rat C6 GBM cells but not in the astrocytes (Figure 1A). The result of Western blotting was in accordance with that of ICC staining, showing the presence of 65 kD IL-13Rα2 in the C6 protein extract (Figure 1B). IL-13Rα2 immunofluorescent staining revealed an undetectable positive green signal in the normal brain tissue (Figure 1C). Distinct IL-13Rα2 immunohistochemical labeling was observed in human GBM cells in the similar staining pattern of C6 cells (Figure 1D).

### 2.2. Sustained Release of Res from Pep-PP@Res

As shown in Figure 2A, after reacting with the sulfhydryl group of Pep-1 peptides, the characteristic peak at 6.7 ppm of the maleimide group (red arrow) disappeared, indicating that Pep-1 peptides was successfully conjugated with a Mal-PEG_3.5K_-*b*-PCL_4K_ copolymer. TEM and DLS demonstrated that the sizes of PP, Pep-PP and PP@Res were less than 30 nm with unimodal distribution (Figure 2B). After being conjugated with Pep-1, the Pep-PP@Res particles maintained unimodal distribution, and their sizes were enlarged to 30 ± 4.8 nm (Figure 2C). The drug loading capacity was 8% for PP@Res and 7.9% for Pep-PP@Res. In vitro Pep-PP@Res release analyses showed a pH-sensitive behavior, and burst release appeared in the first 8 h in both media, especially in the medium at pH 5.0 (Figure 2D). The cumulative Res release from Pep-PP@Res was 75% and 25% at pH 5.0 and pH 7.4 at 48 h, respectively.

### 2.3. Nontoxicity of the Drug Carrier to C6 Cells

The potential cytotoxicity of the drug carrier, PP or Pep-PP, towards C6 cells was evaluated. As shown in Figure 2E, HE staining showed no morphological change of the cells treated by PP and Pep-PP in comparison with that of their normally cultured counterpart. The EdU cell proliferation assay revealed frequent EdU-labeled nuclei in PP- and Pep-PP-treated cell populations (upper left insets in Figure 2E), indicating that the polymers themselves have little influence on cell growth and are nontoxic to C6 cells. 

### 2.4. Pep-1 Modification Enhanced Internalization of Nanoparticles

COU as a harmless lipophilic dye was loaded instead of Res into the nanoparticles and used as a nanoparticle tracer. The cells were treated with 10 ng/mL COU, PP@COU or Pep-PP@COU for 1 h. The intracellular fluorescent intensity was measured and quantified by means of fluorescence microscopy and flow cytometry. Green fluorescence generated by COU was observed in both treated groups. The intensity of fluorescent signals indicated that the nanoparticles were mainly distributed in the cytoplasm. The fluorescent intensity of Pep-PP@COU-treated cells was almost the same as in the COU group and slightly higher than in the PP@COU group (Figure 3A). Further quantitative analysis revealed that the fluorescent intensity of the Pep-PP@COU group remarkably increased (about 1.5-fold) in comparison with that of the PP@COU group and was slightly lower than that of the COU group 1 h after incubation (Figure 3B).

### 2.5. Better Anti-Glioblastoma Efficacy of Pep-PP@Res

The C6 cells were exposed for 4 h to Res, PP@Res and Pep-PP@Res in the concentrations of 20 μM or 100 μM, respectively, and then placed in a normal culture medium. This performance was repeated two times in one-day intervals before the MTT assay. The cell inhibition rate was 55.8 ± 5.5% for 20 μM Pep-PP@Res, compared with 47.2 ± 6.5% for PP@Res (*p* = 0.024) and 45.6 ± 4.0% for free Res (*p* = 0.009) at the same concentration. At the concentration of 100 μM, the differences between the cell inhibition rates of the three Res formulations were narrowed in the form of 75.4 ± 1.4%, 71.9 ± 1.1% (*p* = 0.03) and 71.5 ± 1.1% (*p* = 0.02) for Pep-PP@Res, PP@Res and Res, respectively (Figure 4A). The EdU cell proliferation assay was performed to further demonstrate the growth-suppressive effect of the Res formulations. As shown in Figure 4B, the frequencies of the red EdU fluorescent signal were greatly decreased in the C6 cell populations treated with the three 100 µΜ Res formulations, especially in the Pep-PP@Res-treated one. 

### 2.6. Safety of Pep-PP@Res towards Normal Rat Brain Cells

The 1:2 Matrigel and Neurobasal^TM^-A mixture successfully helped the normal brain tissues to attach to the well surface. Cell outgrowth from the brain tissues was observed after 48 h of culturing and became distinct after one week. The cell-bearing coverslips were then treated with 100 mM Pep-PP@Res for 72 h. As shown in the phase-contrast microscopic images, no obvious morphological change was observed between the brain tissues at the 0 h and 72 h timepoints of the Pep-PP@Res treatment (Figure 5A). The calcein/PI assay showed no signs of cell death (PI-labeled red nucleus) in the cultured brain tissues (the inset in Figure 5A) and brain cells (Figure 5B) after 72 h of the Pep-PP@Res treatment. Immunofluorescent staining further confirmed intact synaptophysin-labeled neurons and GFAP-labeled glial cells in the 2D+ cultured cell population after 72 h of the Pep-PP@Res treatment (Figure 5C).

### 2.7. Prolonged Res Retention Time in Pep-PP@Res-Treated Cells

Res generates green fluorescence at 405 nm that can be captured with fluorescent confocal microscopy [24]. The reason(s) why Pep-PP@Res increased the growth-suppressive rate was investigated by means of fluorescence confocal microscopy-based evaluation of the intracellular retention time of Res. As shown in Figure 6, after 4 h of treatment with Res formulations, granular green fluorescent signals can be observed in whole cells, especially in the nuclei, indicating the general intracellular distribution of Res in the treated C6 cells. The intensity of green fluorescence quickly decreased after withdrawing the Res formulations. At the 24 h timepoint, the green fluorescent signal was barely seen inside the cells, except for the ones treated with Pep-PP@Res.

### 2.8. Improved Inhibitory Effects of Pep-PP@Res on the C6 Xenograft in Nude Mice

The in vivo experimental therapy was conducted in two-day intervals for 12 days with the three Res formulations when the average tumor volume surpassed 1500 mm^3^. As shown in Figure 7A, the tumor-suppressive effect of free Res was weak because of the similar tumor growth rates between the free Res-treated group and the control group; PP@Res and especially Pep-PP@Res exerted a better tumor-inhibitory effect in comparison with that of the control group (*p* = 0.011 and *p* = 0.003, respectively) and the free Res-treated group (*p* = 0.013 and *p* = 0.003, respectively). As shown in Figure 7B, the tumor cells in the control group and the Res-treated group were densely arranged, spindle-shaped, with deep-stained nuclei, a clear nuclear envelope, obvious nucleoli, and the reversed nucleus-to-plasma ratio. In the case of the PP@Res group, the tumor cells became round in shape, with nuclear shrinkage and therefore the decreased nucleus-to-plasma ratio. In the Pep-PP@Res group, the nuclear contraction and fragmentation were more distinct and apoptotic cells were frequently observed, especially in the region around the capillaries. No lymphocyte accumulation or infiltration was observed in the tumor tissues irrespective of Res treatments (Figure 7B).

### 2.9. Activated JNK Signaling in the Res Formulation-Treated C6 Tumors

Because oxidative stress was one of the Res-caused biological events in the cancer cells and JNK signaling is supposed to play an active role by upregulating some proapoptosis genes when cells suffer from oxidative stress [25,26,27], the potential influences of Res in JNK signaling and the expression of JNK-related genes were investigated by means of Western blotting. As shown in Figure 7C, the levels of 46 kD JNK were upregulated 1.28-fold, 2.51-fold and 3.27-fold, respectively, after the treatment with Res, PP@Res and Pep-PP@Res. The level of Bak, a proapoptosis protein, was increased 2.21-fold in the Pep-PP@Res-treated C6 tumors, while it remained largely unchanged in the tumors treated with Res or PP@Res. It was found that the level of Bcl-2, an anti-apoptosis protein, was decreased in the tumors of the PP@Res group (12%) and especially of the Pep-PP@Res group (48%) in comparison with that in the untreated tumors. A little change in Bcl-2 expression was found in the tumors treated with the conventional Res formulation. 

## 3. Discussion

GBM is a lethal primary brain malignancy with high recurrence and mortality rates due to its aggressive growth and the difficulty of radical resection [6]. Adjuvant chemotherapy is therefore required to reduce the risk of tumor relapse [5]. As a natural polyphenolic compound, Res exerts inhibitory effects on GBM cells without affecting normal tissues and cells [12,28], suggesting its potential therapeutic value for GBMs [29,30]. Nevertheless, the pharmacokinetic and physicochemical drawbacks of Res greatly limit its in vivo application when administered in the form of free molecules [13,28,31]. Nanoparticles (NPs) have been used as a drug carrier to encapsulate Res molecules, and the prepared reagent shows controlled release [15,31]. The active targeting function of nanoparticles is to connect the nanoparticles with proteins or peptides that have a targeting effect so that the particles have the role of targeting specific cells. IL-13Rα2, a transmembrane protein of the IL-13R family, actively mediates endocytosis [20]. More importantly, IL-13Rα2 is absent or expressed at a very low level in normal brain tissues but is overexpressed in GBMs [21] as well as in GBM cell lines as demonstrated in this study. Therefore, IL-13Rα2 is regarded as a drug delivery target for GBM treatment [19,21]. For these reasons, we modified NPs with a peptide (Pep-1) proved to be the corresponding ligand of IL-13Rα2 for GBMs treatment [23].

As the first experimental step, resveratrol was successfully encapsulated into NPs and exhibited less than 50 nm with unimodal distribution after modification with Pep-1, suggesting its advantage to reduce the kidney and reticuloendothelial system (RES) removal [32]. According to the literature, Res reaches the highest concentration within the first 5 min in the cytoplasm but has a very short half-life (e.g., 14.4 min in rabbits) when administered systemically [30]. To overcome this therapeutic bottleneck, it is necessary to protect Res molecules from being metabolized and eliminated before they reach the target sites. As the next step of our investigation, the Res release pattern of Pep-PP@Res was evaluated. We found that Pep-PP@Res not only exhibited sustained Res release, but also displayed the acid response property. These findings suggest the substantial benefits of Pep-PP@Res for practical applications because this formulation can avoid metabolization of the circulating Res and thus increase the Res bioavailability in mild acidic tumor growth sites [33,34]. 

Safety is the precedent condition of practical use of a drug formulation. Before testing the anti-GBM effect of the Res formulations, cytological staining was performed on the C6 cells treated with Res-free PP and Pep-PP, respectively. The results showed that the growth rates of the PP- and Pep-PP-treated cell populations were similar to those of their normally cultured counterparts, suggesting little cytotoxic effect and the suitability of these two drug carriers for packaging Res to generate GBM-targeting Pep-PP@Res. Alternatively, it would be reasonable to consider that the anti-GBM effects caused by Pep-PP@Res are the consequence of multifaceted actions of Res. The influence on the growth and survival of normal brain cells is another critical point of drug safety. To address this issue and provide preexperimental data for the treatment of the C6-formed rat orthotopic xenograft with Pep-PP@Res, we cultured the microdissected rat brain tissues under the two-dimensional plus (2D+) condition by covering the tissue slices with the Matrigel/Neurobasal^TM^-A mixture. The outgrowth of brain cell components was observed at day 2 and became active one week after the culturing. To guarantee the reliability of safety evaluation, a high concentration (100 μM) of Pep-PP@Res was adopted to treat the brain tissue-generated cells. Multiple examinations demonstrated that a powerful anti-GBM concentration of Pep-PP@Res exerted little unfavorable effect on the normal brain cells including the GFAP-positive glial cells and synaptophysin-expressing neurons. The results above confirm (1) that the anti-GBM effects of Pep-PP@Res resulted from Res rather than PP or Pep-PP and (2) that Pep-PP@Res are able to suppress GBM cells without affecting normal brain cells.

The GBM-oriented targeting of the Pep-1 peptide is another key point to be addressed. Because Res can cause GBM cell death, coumarin (COU) was employed as a fluorescent tracer in the qualitative and quantitative analyses of Pep-1-targeting efficacy. The results revealed that the Pep-PP@COU-treated C6 cells showed stronger fluorescent intensity than those treated with PP@COU. It has been known that resveratrol and COU are fat-soluble drugs that mainly enter the cell through passive diffusion, while NPs, especially NPs with the tumor cell membrane receptor-targeting function, rely on more efficient receptors to mediate endocytosis into cells [35,36,37]. The stronger fluorescent intensity in the Pep-PP@COU-treated C6 cells may be the consequence of the Pep-1-enhanced IL-13Rα2-mediated endocytosis, resulting in the increased uptake of Pep-PP@Res and, consequently, stronger anti-GBM efficacy. It has been recognized that the intracellular Res concentration is very low and its retention time is extremely short when it is administered systemically [30]. To imitate the actual in vivo situation, the C6 cells were exposed to 20 μM and 100 μM Res for 4 h/day to determine whether the anti-glioblastoma efficacy can be improved by the use of Pep-PP@Res. Better inhibitory outcomes were achieved with both short-term 20 μM and 100 μM Pep-PP@Res exposure than with free Res and PP@Res treatments. Thus, these results further support the GBM-targeting property of Pep-1 as well as its ability to enhance cell uptake. To further confirm the practical values of Pep-PP@Res, we investigated its blood–brain barrier permeability and brain bioavailability in a normal rat and its orthotopic glioblastoma model.

As the final goal of this study, the in vivo anti-glioblastoma efficacy of Pep-PP@Res was tested by systemically administering PP@Res or Pep-PP@Res to the nude mice bearing C6-formed subcutaneous transplanted tumors six times in two-day intervals. By the end of the treatment, 47.3% reduction of the relative volume of the C6 xenograft tumor was observed in the nude mice treated with PP@Res and a more distinct reduction (64.5%) in the Pep-PP@Res group in comparison with that of the control group. This notable suppression of tumor growth was attributed to the use of the nanocapsules as a delivery system because the tumors in the free Res-treated mice kept growing at a similar rate to that of the animals without treatment. It is therefore considered that PP@Res and especially Pep-PP@Res have a promising in vivo anti-glioblastoma effect. It should be pointed out that Pep-PP@Res only reduce the growth rate of transplanted tumors but fail to eliminate the tumors radically. It would be worthwhile to elucidate whether the in vivo GBM suppression efficacy can be further improved by more frequent administration such as daily instead of once per two days, earlier initiation of the treatment when tumors are smaller (30–50 mm^3^) and/or intravenous injection instead of an intraperitoneal one [38]. Reservatrol has been known as an important modulator of the immune response [39], while no signs of lymphocyte accumulation and infiltration were observed in the tumor regions after the Res treatments. Because of the immunodeficiency of the nude mice, it is necessary to address the presence of Res-caused immunological response using a rat orthotopic glioblastoma model.

N-terminal kinases (JNK) are the main family members of MAP kinases, which play pivotal roles in promoting proliferation, differentiation and apoptosis [40]. JNK signaling becomes activated when the cells are under oxidative stress, leading to the upregulation of some proapoptosis genes [26,27,41]. Because oxidative stress is one of the biological events in Res-treated cancer cells including GBM cells [25,42], the relevance of JNK and Bak, as well as of Bcl-2 in Res-treated C6 tumors was evaluated. The results revealed that JNK levels were distinctly increased after the Pep-PP@Res treatment, accompanied by increased Bak and reduced Bcl-2 production, confirming from another angle the activating effect of Res on JNK signaling. It has been known that Bak can increase the mitochondrial voltage-dependent anion channel (VDAC), leading to the loss of the membrane potential, the release of cytochrome c [43] and cascade activation of caspases [44]. Our previous study demonstrated that Res is able to cause oxidative damage in Res-sensitive GBM cells, leading to mitochondrial structural alterations and caspase-9 and -3 activation [42]. The results of this study further suggest that the oxidative stress induced by Res may result in apoptosis through activating JNK signaling and, therefore, reversing Bak and Bcl-2 expression patterns.

## 4. Materials and Methods

### 4.1. Experimental Materials

Resveratrol, dimethyl sulfoxide (DMSO) and coumarin-6 (COU) were purchased from Sigma-Aldrich Co. (St. Louis, MO, USA). The Pep-1 peptide with the sequence “CGEMGWVRC” was purchased from ChinaPeptides Co., Ltd. (Shanghai, China). PEG_2K_-*b*-PCL_4K_ and Mal-PEG_3.5K_-*b*-PCL_4K_ were synthesized as described before [44,45,46]. MTT (3-(4,5-dimethyl-2-thiazolyl)-2,5-diphenyl-2-H-tetrazolium bromide) was purchased from Futong Biotech. Inc. (Guangzhou, China). The BeyoClick^TM^ EdU Cell Proliferation Kit with Alexa Fluor 594, the BCA protein assay kit, the JNK antibody and the p-JNK antibody were purchased from Beyotime Institute of Biotech. Inc. (Beijing, China). The immunohistochemical staining kit for *Streptomyces* anti-biotin protein–peroxidase (SP kit) was purchased from Zsgb Institute of Biotech. Inc. (Beijing, China). The IL-13Rα2 antibody, the β-actin antibody, the goat anti-mouse IgG, the goat anti-rabbit IgG, the Coralite488-conjugated goat anti-rabbit IgG and the Bak antibody were provided by Abcam Biotech. Inc. (Cambridge, UK). The Bcl-2 antibody was provided by Bioss Biotech. Inc. (Beijing, China). The Matrigel matrix was purchased from Corning Inc. (Corning, NY, USA), the Neurobasal^TM^-A medium—from ThermoFisher Scientific Inc. (Waltham, MA, USA). The Calcein/PI cell viability assay was the product of Beyotime Inst Biotech. Inc. (Beijing, China).

### 4.2. GBM Cell Line 

The rat C6 glioblastoma cell line was generously provided by the Central Laboratory of the Shanghai Tenth People’s Hospital [47]. The cells were cultured in DMEM with H-glutamine (Gibco; Thermo Fisher Scientific, Waltham, MA, USA), supplemented with 10% fetal bovine serum (Gibco Life Science, Grand Island, NY, USA) and 50 U/mL penicillin and 50 μg/ml streptomycin (Gibco Life Science) in a stable atmosphere of 5% CO_2_ at 37 °C.

### 4.3. Immunocytochemical and Immunofluorescent Staining

Immunocytochemical staining (ICC) was employed to check IL-13Rα2 expression in the rat C6 glioblastoma cells. Briefly, the GBM cells (2 × 10^5^) were grown on the coverslips placed in a culture dish until about 70% confluence. The cell-bearing coverslips were harvested and washed with a phosphate-buffered solution (PBS, pH 7.4). After being incubated for 10 min in 3% H_2_O_2_ and then with goat serum blocked at 37 ℃ for 20 min, the coverslips were incubated with the IL-13Rα2 (1:500) antibody at 4 ℃ overnight in a humid chamber, followed by the treatments with the biotin-labeled goat anti-mouse/rabbit IgG at room temperature for 30 min and with the polymerized horseradish peroxidase (HRP)-labeled streptomycin working fluid at 37 ℃ for 15 min. A color reaction was developed using 3,3′-diaminobenzidine tetrahydrochloride (DAB). Immunofluorescent staining (IF) was adopted to investigate the expression of IL-13Rα2 in normal brains of mice. The BALB/c mice (weighing 20 ± 5 g) were housed under SPF conditions. After euthanasia, the brains were removed and fixed in 4% paraformaldehyde, embedded into paraffin and sectioned for IL-13Rα2-oriented immunofluorescent staining. 

### 4.4. Synthesis of Pep-PEG3.5K-b-PCL4K

The Pep-1 peptides comprising nine amino acids (CGEMGWVRC) in the amount of 25 mg were dissolved in 1 mL phosphate buffer (pH 7.4, 10 mM) and treated with nitrogen atmosphere for 30 min. In the meantime, 20 mg PP (Mal-PEG_3.5K_-*b*-PCL_4K_) was dissolved in 1 mL dimethylformamide, mixed with 2 mL phosphate buffer solution (pH 7.4, 10 mM) and stirred for 1 h. Then, these two solutions were mixed and stirred for 12 h at room temperature under nitrogen atmosphere. Lastly, the polymer was dialyzed with ultrapure water for 2 days, freeze-dried and characterized by ^1^H NMR (yield: 87.2%).

### 4.5. Preparation of Nanoparticle Formulations

Pep-1-conjugated Res-loaded PEG_2K_-*b*-PCL_4K_ nanoparticles (Pep-PP@Res) were constructed using the method described in our recent publication [48]. Briefly, PP (PEG_2K_-*b*-PCL_4K_) and Res at the weight ratio of 5:1 were co-dissolved in acetone and stirred for 30 min to make sure the materials dissolved completely. Ultrapure water was added drop-wise, and the mixture stirred for 1 h at room temperature. Acetone was removed under vacuum. The residual solution was centrifuged at 3000 rpm for 10 min to remove the free drugs. As for the Pep-1-conjugated Res-loaded nanoparticles (Pep-PP@Res), the weight ratio of Pep-PEG_3.5K_-*b*-PCL_4K_, PEG_2K_-*b*-PCL_4K_ and Res was 1:9:2, and other experimental procedures were the same as PP@Res preparation. The empty nanoparticles (PEG_3.5K_-*b*-PCL_4K_, PP; Pep-PEG_3.5K_-*b*-PCL_4K_, Pep-PP) and COU-loaded formulations were prepared according to the procedures similar to those for the Res-loaded nanoparticles. 

### 4.6. Characterization of the Res-Loaded Formulations

The morphology and size distribution of Res nanoparticles were carried out using transmission electron microscopy (TEM; JEM-1400plus, JEOL, Japan) and dynamic light scattering (DLS) (ZEN1690, Malvern, UK), respectively. The drug-loading content (DLC) of PP@Res and Pep-PP@Res was measured by means of ultraviolet spectrophotometry (Evolution 300, Thermo Fisher Scientific, Waltham, MA, USA). The calibration curve was linearized in the range of 1–5 μg/mL with the correlation coefficient of R^2^ = 0.9999. To ascertain the amount of Res encapsulated in the nanoparticles, a mixed solution of DMSO and ultrapure water (1:1, v/v) was used to dissolve the particles. DLC% was calculated as indicated below (*n* ≥ 3):(1)$$\mathrm{DLC}\%=\frac{\mathrm{Amount}\;\mathrm{of}\;\mathrm{Res}\;\mathrm{in}\;\mathrm{the}\;\mathrm{nanoparticles}}{\mathrm{Total}\;\mathrm{weight}\;\mathrm{of}\;\mathrm{the}\;\mathrm{Res}-\mathrm{loaded}\;\mathrm{nanoparticles}\;}\times 100\%$$


### 4.7. In Vitro Drug Release Analysis

The in vitro release of Res from the nanoparticles was performed in a PBS (pH 5.0 and 7.4) by means of dialysis at 37 ℃ under horizontal shaking at 100 rpm. Briefly, the Pep-PP@Res nanoparticles (containing 158.3 µg of Res) were dissolved in 100 μL PBS, placed into a dialysis bag (MWCO 3500) and dipped in 20 mL PBS. Five milliliters of the PBS were withdrawn from the external medium after 0.5, 1, 2, 4, 8, 24, 48, 72 and 96 h of incubation and replaced with 5 mL fresh medium. The samples withdrawn from the external medium were lyophilized and dissolved in 1 mL mixed solution (DMSO/water = 1:1, v/v) and analyzed by means of ultraviolet spectrophotometry as described above.

### 4.8. Drug Carrier Safety Study

To check the cytotoxicity of the Res-free nanoparticles, the C6 cells treated with 115 μg PP or Pep-PP were subjected to hematoxylin and eosin morphological staining (HE staining). Meanwhile, the C6 cells (2.5 × 10^4^) were cultured on coverslips placed in 12-well plates, treated with PP and Pep-PP, respectively, for 48 h and then fixed with 4% paraformaldehyde for the EdU cell proliferation assay by the method described elsewhere [48]. Briefly, the cell-bearing coverslips of each of the experimental groups were labeled with 5-ethynyl-2′-deoxyuridine (EdU) for 2 h, then incubated with Click Additive Solution at room temperature for 30 min, followed by staining the cell nuclei with Hoechst for 10 min in darkness. The cell images were collected under fluorescence microscopy (Imager M2, Zeiss, Germany)

### 4.9. Internalization of Pep-PP@COU into GBM Cells 

For qualitative analysis, the C6 cells (2.5 × 10^4^) were grown on coverslips placed in 12-well plates until about 70% confluence. The cells were treated with COU, PP@COU and Pep-PP@COU at the final lipid concentration of 10 ng/ml. After 1 h incubation, the cells were washed three times with an ice-cold PBS and fixed with 4% paraformaldehyde for 10 min, followed by nuclei staining with DAPI for 10 min and then ice-cold PBS washing (three times); the fluorescent images were collected under fluorescence microscopy (Imager M2, Zeiss, Germany). For quantitative analysis, 2.5 × 10^4^ C6 cells were seeded in 12-well plates until about 70% confluence. The cells were treated with COU, PP@COU and Pep-PP@COU at the final lipid concentration of 10 ng/mL. After 1 h incubation, the cells were washed three times with cold PBS and detached using 0.25% trypsin-EDTA. The cell pellet was collected by means of 800 rpm centrifugation for 5 min and rinsed three times with an ice-cold PBS. The cell pellets were resuspended in 1 mL ice-cold PBS on ice for flow cytometry at an excitation wavelength of 488 nm. The emission of COU was recorded in the FL-1 channel, and the average fluorescent intensity in each group was collected. 

### 4.10. Cell Proliferation Assay

Short-term (4 h/day) drug treatment was conducted by exposing the cells to 20 μM or 100 μM Res, PP@Res and Pep-PP@Res for 4 h, and then the drug-containing medium was replaced with the normal culture medium until the next 4 h drug treatment 20 h later. After repeating the short-term treatments twice, the cells were incubated with 10 μL MTT/well for 4 h. After removing the supernatants, formazan was dissolved in 100 μL DMSO and analyzed using a Microplate Reader (Varioskan LUX, ThermoFisher Scientific, USA) [9]. In parallel, the cell-bearing coverslips were collected in 100 μM groups, labeled with 5-ethynyl-2′-deoxyuridine (EdU) in a cell incubator for 2 h and fixed with 4% paraformaldehyde for 10 min, followed by 0.3% Triton X-100 treatment for 10 min. After washing three times with 3% BSA, Click Additive Solution was added onto each coverslip and incubated in darkness at room temperature for 30 min. The cell nuclei were stained with Hoechst 33342 in darkness for 10 min and analyzed by means of fluorescence microscopy (Imager M2, Zeiss, Jena, Germany). Each of the experimental groups was set in triplicate wells, and 10–15 images (x 10) were obtained from each of the wells and subjected to data collection. The experiment was repeated three times to establish a reliable conclusion.

### 4.11. Safety of Pep-PP@Res on Normal Rat Brain Cells

Because the C6 cell line was derived from a rat glioblastoma [47] and Pep-PP@Res was used to treat a rat orthotopic glioblastoma model, the brain tissue of a normal rat was employed to elucidate the neurological safety of the Pep-PP@Res particles. Briefly, the fresh brain tissue was mechanically minced to micro-pieces and centrifuged at 300 rpm for 5 min to collect the pellets. The pellets were seeded on a panel of coverslips and then coated with a thin layer of a Neurobasal^TM^-A-containing Matrigel matrix (1:2). The coverslips were put into a 48-well plate (one coverslip/well) and cultured in 200 mL Neurobasal^TM^-A medium. When distinct cell outgrowth was observed, the cell-bearing coverslips were treated with 100 mM Pep-PP@Res for 72 h. The phenotypes of brain cells before (0 h) and after the 72 h drug treatment were imaged daily. The status of cell death was examined at the 0 h and 72 h timepoints of the Pep-PP@Res treatment using the calcein/PI cell viability assay. The cultured brain cells were subtyped by means of immunofluorescent labeling with rabbit anti-human GFAP (Beyotime Biotech. Inc. Shanghai, China; 1:250) and a rabbit anti-synaptophysin antibody (Beyotime Biotech. Inc. Shanghai, China; 1:200) using the method described elsewhere [12].

### 4.12. Intracellular Res Retention Time and Distribution Pattern

In order to investigate the retention time and intracellular location of Res, the C6 cells were seeded on cell coverslips and treated with the Res formulations for 4 h; the culture media were replaced by complete DMEM for normal culturing. After normal culturing for 0, 1, 4 and 24 h, the cell coverslips were collected, fixed with 4% paraformaldehyde for 30 min, and the fluorescence of Res was observed and photographed under a fluorescence confocal microscope (LSM800, Zeiss, Jena, Germany). The images were acquired at 488 nm (FITC filter).

### 4.13. Evaluation of In Vivo Tumor-Suppressive Effects

All the animal experiments were carried out in accordance with the animal protocol approved by the ethics committee of South China University of Technology (SCUT; AEC No. 2018050). In this study, 4–6-week-old male BALB/c nude mice (20 ± 5 g) were purchased from Hunan SJA Laboratory Animal Co., Ltd, and housed in SPF conditions. To establish a subcutaneous glioblastoma model, 5 × 10^6^ cells were injected to each of the inoculation sites, and 23 transplanted tumors were formed. Once the tumors grew to 50–150 mm^3^, the nude mice were randomized to the normal saline-treated, free Res-treated, PP@Res-treated and Pep-PP@Res-treated groups in the numbers of 3, 4, 8 and 8, respectively. The dose of Res in the free form or encapsulated in the nanoparticles was 30 mg/kg. The formulations were intraperitoneally administered (i.p.) to the mice six times in two-day intervals. The tumor volume and the relative tumor volume were calculated as shown below, where *W* is the widest point and *L* is the longest point. The in vivo study was terminated when the average tumor volume surpassed 1500 mm^3^. At the end of the treatment, the mice were euthanized and the tumors were removed; portions were fixed with 4% paraformaldehyde and embedded into paraffin and sectioned for morphological examination; the others were kept at −80 ℃ for protein preparation.
(2)$$\mathrm{Tumor}\;\mathrm{volume}=\left(W\times W\times L\right)/2$$



(3)
$$\mathrm{Relative}\;\mathrm{tumor}\;\mathrm{volume}=\frac{\mathrm{Volume}\;\mathrm{measured}\;\mathrm{at}\;\mathrm{the}\;\mathrm{corresponding}\;\mathrm{day}}{\mathrm{Volume}\;\mathrm{measured}\;\mathrm{at}\;\mathrm{day}\;0}$$



### 4.14. Protein Preparation and Western Blotting

Total cellular proteins were prepared from the cells under different culturing conditions. Each of the sample proteins (30 µg) was added into the well and separated by means of 10% polyacrylamide gel electrophoresis (Bio-Rad, Hercules, CA, USA) and transferred to a polyvinylidene difluoride membrane (Bio-Rad, Hercules, CA, USA). The membrane was blocked by 5% skimmed milk in TBS-T (10 mM TrisCl, pH 8.0, 150 mM NaCl, 0.5% Tween-20) at room temperature for 2 h, rinsed gently with TBS-T, followed by incubation with the first antibody in appropriate concentrations (IL-13Rα2: 1:800, p-MKK7: 1:500, JNK: 1:800, p-JNK: 1:500, c-Jun: 1:800, Bak: 1:500, Bcl-2: 1:500, β-actin: 1:10,000) at 4 ℃ overnight, followed by 1 h incubation with HRP-conjugated goat anti-rabbit or anti-mouse IgG. Each step in the reaction with the antibody was followed by 10 min rinsing with TBS-T (three times). The bound antibody was detected using Amersham Imager 600 series imagers (GE Healthcare, Chicago, IL, USA). Each membrane was washed three times with TBS-T, blocked by 5% skimmed milk in TBS-T for 2 h and reused at least twice to identify different kinds of proteins or preserved at –20 °C.

### 4.15. Statistical Analysis

All the experiments were repeated at least three times, and statistical analysis was performed with the SPSS 13.0 software. An analysis of variance (ANOVA) followed by Tukey’s post-hoc test or Student’s *t*-test was performed to evaluate statistical significance. All the numerical values are presented as the means ± SD; *p* < 0.05 was considered for statistical significance.

## 5. Conclusions

The results of this study demonstrate that Pep-PP@Res significantly prolong the Res release time in vitro and the intracellular Res retention time. Pep-PP@Res exert more efficient growth suppression effects on C6 cells. The in vivo anti-glioblastoma effects of Res can be distinctly improved in the form of Pep-PP@Res nanoparticles via activating JNK signaling, upregulating proapoptosis gene expression, finally resulting in extensive apoptosis. Moreover, Pep-PP@Res are nontoxic to normal brain cells. To our knowledge, this is the first report on the correlation of JNK signaling and Res in an experimental GBM system. Pep-PP@Res would be of practical value for better management of GBMs because of their sustained release and GBM-targeting properties. Further investigations should be conducted to elucidate the in vivo pharmacokinetic features and immunomodulating activity of Pep-PP@Res using a C6-formed rat orthotopic model.

## Figures and Tables

**Figure 1 ijms-22-10622-f001:**
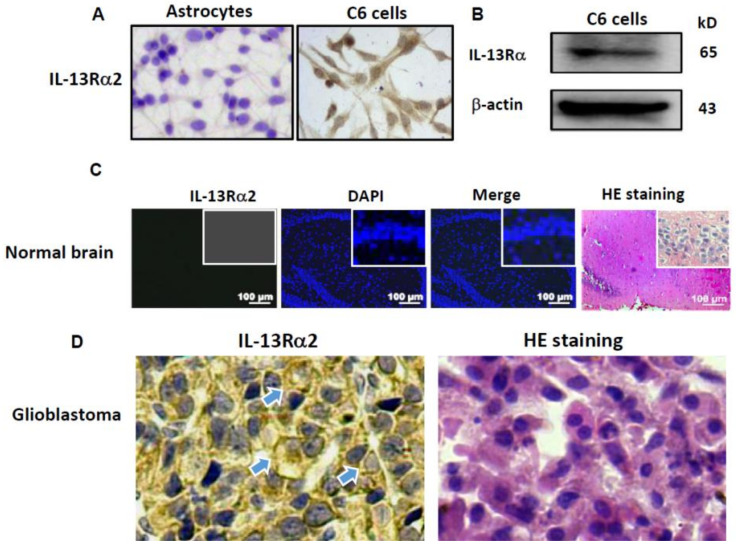
Distinct IL-13Rα2 immunohistochemical staining in rat C6 glioblastoma cells in vitro and human glioblastoma cells in vivo but not in rat astrocytes and normal brain tissue. (**A**) IL-13Rα2-oriented immunocytochemical staining performed on rat astrocytes and C6 cells; (**B**) Western blotting demonstration of IL-13Rα2 expression in C6 cells; (**C**) absence of IL-13Rα2 immunofluorescent labeling in the normal rat brain tissue; (**D**) immunohistochemical demonstration of IL-13Rα2 expression and membranous location (arrow-indicated; ×40) in the case of human glioblastoma multiforme.

**Figure 2 ijms-22-10622-f002:**
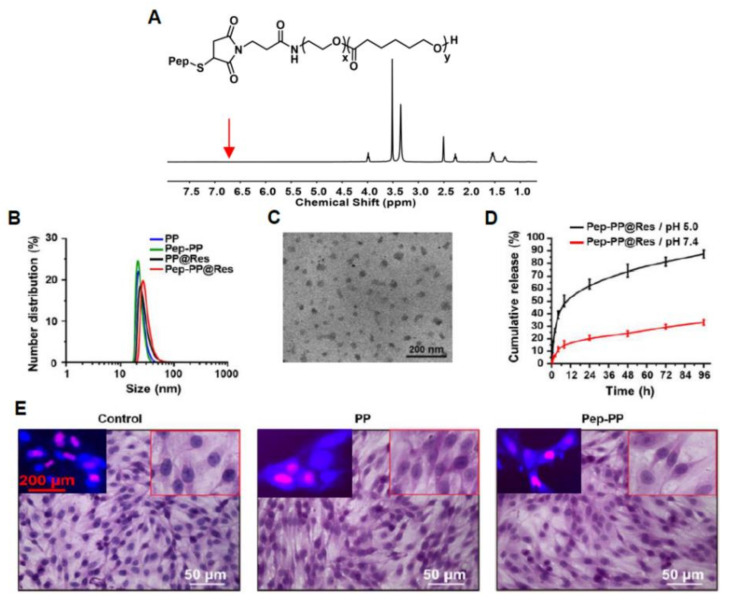
Construction and characterization of encapsulated resveratrol and IL-13Rα2-targeted sustained-release nanoparticles Pep-PP@Res. (**A**) ^1^H NMR spectra of Pep-PEG_3.5K_-*b*-PCL_4K_ in DMSO-*D*_6_; (**B**) size distribution of nanoparticle formulations through DLS; (**C**) morphology of Pep-PP@Res under TEM; (**D**) in vitro release curves of Pep-PP@Res under different pH conditions; (**E**) demonstration of the nontoxic effect of PP and Pep-PP on C6 cells by HE morphological staining and the EdU cell proliferation assay (upper left insets). Scale bars, 50 μm in the main images and 200 μm in the insets.

**Figure 3 ijms-22-10622-f003:**
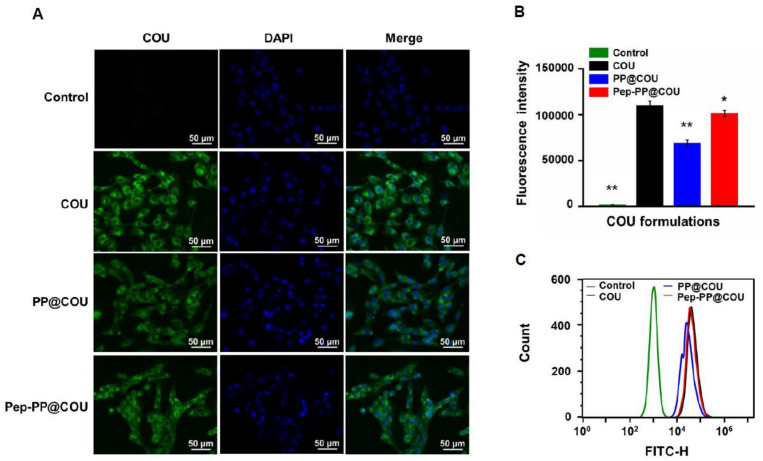
Evaluation of nanoparticle internalization efficacy via quantification of cellular uptake of COU, PP@COU and Pep-PP-COU. Cellular uptake of COU, PP@COU and Pep-PP-COU was illustrated (**A**) and quantified (**B**) through tracing COU fluorescent intensities. Scale bar, 50 μm. Quantitative analysis of cellular uptake using flow cytometry (**C**). Note: ***** and ****** indicate *p*-values < 0.05 and < 0.001, respectively, when compared with Pep-PP@COU.

**Figure 4 ijms-22-10622-f004:**
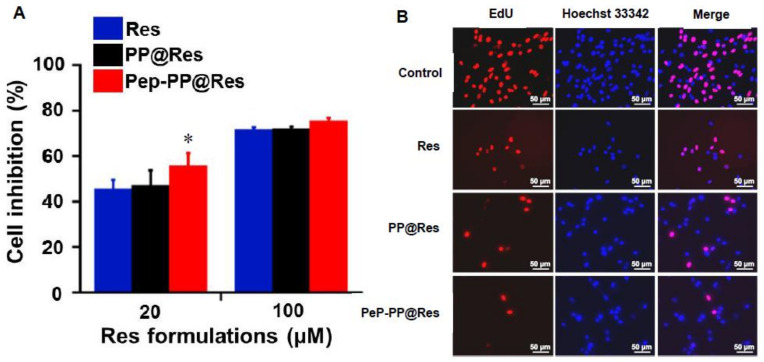
Growth-suppressive effects of the Pep-PP@Res nanoparticles on the C6 cells. (**A**) The cell proliferation assay showed distinct cell inhibition of the C6 cells after three repeats of short-term (4 h) treatments with the three Res formulations. Note: ***** indicates *p*-values < 0.05 when compared with Res and PP@Res. (**B**) Reduction of EdU-labeling frequencies in the C6 populations treated with Res, PP@Res and Pep-PP@Res. Red signals, EdU-labeled nuclei of proliferating cells; blue signal, Hoechst-labeled nuclei of total cells; merged pink signals, the nuclei of proliferating cells.

**Figure 5 ijms-22-10622-f005:**
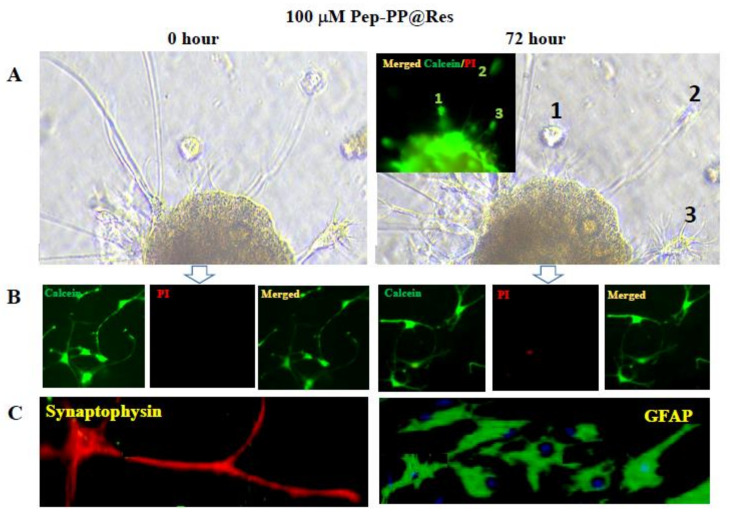
Pep-PP@Res exerted little toxic effect on the 2D+ cultured normal rat brain cells. (**A**). Bright field illustration of the normal rat brain tissue with cell outgrowth and without cell death (the inset) after 72 h of the 100 μM Pep-PP@Res treatment. The cells marked with 1, 2 and 3 in the bright field image and the inset are the same viable cells without PI labeling. (**B**). The Calcein/PI cell viability assay (viable (green)/nonviable (red)) performed on the rat brain cells at the 0 h and 72 h timepoints of the Pep-PP@Res treatment, demonstrating the rarity of brain cell death. (**C**). Immunofluorescent illustration of the intact synaptophysin-positive neurons and GFAP-positive glial cells in the 2D+ brain tissue culture treated with 100 μM Pep-PP@Res for 72 h.

**Figure 6 ijms-22-10622-f006:**
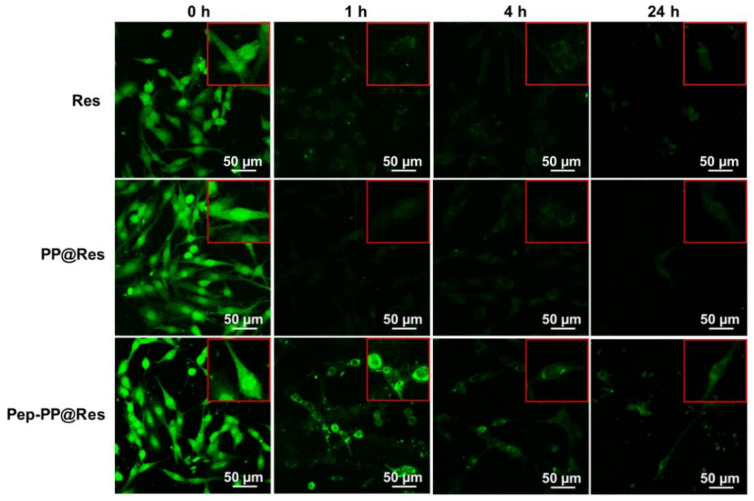
Distribution pattern and different intracellular retention times of the three Res formulations. Scale bars, 50 μm in the main images and 100 μm in the insets.

**Figure 7 ijms-22-10622-f007:**
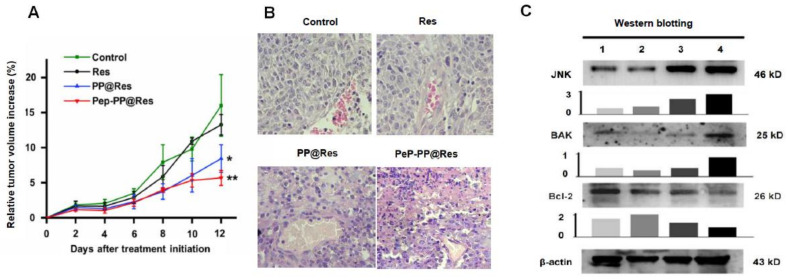
Pep-PP@Res inhibited growth and activated JNK signaling of the C6 xenografts in nude mice. (**A**) Relative tumor volume increase (%) after drug treatments. Note: ***** and ** indicate *p*-values < 0.05 and < 0.01 in comparison with the control group. (**B**) The morphology of tumor tissues in the four experimental groups and the extensive cell death in the Pep-PP@Res-treated tumor tissues; scale bar, 50 μm. (**C**) Western blot examination of JNK signaling statuses in the C6 cells without drug treatment (1) and treated by free Res (2), PP@Res (3) and Pep-PP@Res (4). Grayscale analyses were conducted on the Western blot results.

## Data Availability

Not applicable.

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
