# Peer review of "Construction of IL-13 Receptor α2-Targeting Resveratrol Nanoparticles against Glioblastoma Cells: Therapeutic Efficacy and Molecular Effects"

_ijms, 2021, doi:10.3390/ijms221910622_

Round 1

Reviewer 1 Report

The authors have responded to all concerns of the reviewers.

Author Response

Thank you very much for your critical reading and previous valuable comments.

Reviewer 2 Report

In this manuscript, Lin et al. describe the development of a resveratrol nanocarrier to glioma treatment. The authors show that encapsulation of resveratrol into a polymeric nanoparticle targeting IL-13R alpha2 increased its cell concentration triggering the anti-glioblastoma effect by inducing JNK signaling activation leading to apoptosis. The topic is relevant; however, several flaws limit the enthusiastic of the manuscript. Additionally, several papers in the literature are showing increased brain delivery of resveratrol through several nanoparticles.
1. The manuscript is barely written and an extensive revision is a need. Figure legends must be improved so the reader clearly understands the result presented.
2. Methods must be re-written including important information, such as the method to determine glioma growth.
3. Nanoparticle preparation and characterization description must be improved. It was developed a solid nanoparticle? After preparation, the formulation was centrifuged to remove free drugs. Was the formulation resuspended (in which solution?) after free drugs are removed? Data about zeta potential and formulation stability should be included as well.
4. Why do the authors chose to evaluate the internalization of coumarin-loaded nanoparticles as a proxy of resveratrol-loaded nanoparticle internalization?
5. Authors use a glioma cell line-derived of rat, performed some experiments using nude mice, others using rats making it difficult to conclude some results.
6. Figure 1, negative control by not incubating the slide with a primary antibody not means the IL-13Ra2 is not expressed. The authors must compare glioma cells with astrocytes. In the same way, Coralite 488 stains IL-13Ra2 making this experiment no sense.
7. Additional experiments should be performed to evaluate the toxicity of nanoparticles
8. Internalization of coumarin by nanoparticle was worse of coumarin free (fig. 3B).
9. Treatments of cells are not clear. Further, there is a misconception about dose and concentration concepts throughout the manuscript.
10. Cell inhibition induced was not improved by resveratrol-loaded nanoparticles, mainly at 100 uM concentration.
11. Safety of resveratrol-loaded nanoparticles to brain cells experiment is hard to understand. Authors must consider another way to evaluate, including markers of neurons, glial cells...
12. I didn't understand the graphic in figure 7C. What does mean 0-2, 0-1, 0-3 in the y axis?

Author Response

Many thanks for your valuable comments, attached p[lease find our response.

Reviewer 3 Report

In this paper, Lin and colleagues have reported the construction of Pep-PP@Res (Res-encapsulated IL-13Ra2 targeting nanoparticles) and shown the efficacy in the prolonged Resveratrol (Res) release of Pep-PP@Res and its killing activity against GBM via JNK activation. This manuscript is well written and data sound interesting. I have some comments.

The authors prepared Pep-PP@Res formulations and applied them to in-vitro and in-vivo targeting and killing GBM in this study. But it is somewhat hard to conceptualize the methodology as well as terminology used in formulating the constructs such as Pep-PP@Res or PP@Res. It is recommended to include a Figure or flow chart to summarize the entire processes and materials in preparing its formulation, which would be beneficial for general readership.

Figure 1 shows the results of IL-13Ra2 expression in C6 cell line. Western blot results in B support immunocytochemical staining results (A). It is nice if the original full blot data are included as the Supplementary Data. Immunofluorescent analysis shows no or less expression of IL-13Ra2 in normal mouse brain tissues. It will be nice if any positive in-vivo data (e.g., GBM tissues) are included if possible.

Figure 4 indicates the effects of Pep-PP@Res on killing C6 cells. I am curious of the killing effect of Pep-PP@Res to C6 cells compared with other constructs. Was Pep-PP@Res really significantly increasing the killing activity to C6 especially at 100 microM? How many images or experiments were used to do these comparisons? The asterisks should be put on the red bars (Pep-PP@Res) after comparison with other control constructs.

Figure 7A shows the better inhibition of Pep-PP@Res to xenograft tumor growth. What is the unit relative tumor volume? It is not clear. The authors should modify the Y axis of the graph by using evident units (e.g., mm3).

Author Response

Many thanks for your valuable comments. Please find our response in the attachment.

Round 2

Reviewer 2 Report

The authors have addressed my concerns and the paper was significantly improved. However, I have some questions:
1. Are the authors sure they used the anti-synaptophysin antibody in figure 5C? Given that synaptophysin is a pre-synaptic protein, immunofluorescence images show stain puncta and the neuron in the figure is whole stained. It appears they used MAP2 antibody instead.
2. The question about the use of nude mice remains unanswered. Why didn't the authors injected C6 cells into the brain of rats? Additionally, tumor microenvironment is important to tumor growth and malignance and play a central role in therapeutic strategies.

Author Response

The authors highly appreciate your comments that are of values in improving the quality of current manuscript and our future works, many thanks. Attached please find our response to your comments.

Reviewer 3 Report

Thank you for the revision. Some of my concerns have been addressed in the current revision which made this manuscript improved. I have comments related to the revised Figure 1.

  • A (immunostaining of cells): The authors showed the results of IL-13Ra2 expression in astrocytes and C6 cells. Are the astrocytes isolated from brain tissues of rats? If so, please define and provide the method of isolating them for general readership, if it hasn’t been yet included in the revised manuscript. Are the astrocytes used as a negative control?
  • B (immunoblot): The authors provided the full-sized blots in the supplementary material. They should be the original images generating the cropped ones shown in the Figure 1B. In addition, it is not clear which bands correspond to the IL-13Ra2 in the newly added supplementary full blot.
  • D (immunostaining of tissue section): The authors included the new immunostaining data in the revised Figure 1. The arrows might be indicating IL-13Ra2 expressions in the glioblastoma tissue. To my knowledge, it is somewhat difficult to recognize which signals are specific for IL13Ra2. Also, it would be nice if the results from any negative control are provided if available.

Author Response

The authors highly appreciate your comments that are valuable for improving the quality of our current manuscript and future works, many thanks!

Round 3

Reviewer 3 Report

The authors addressed my concerns.

This manuscript is a resubmission of an earlier submission. The following is a list of the peer review reports and author responses from that submission.

Round 1

Reviewer 1 Report

The work by Xiao-Min Lin and colleagues describes the development and study of glioblastoma directed nanoparticles encapsulating resveratrol, a pleiotropic polyphenol with acknowledge anti-tumor effects. Resveratrol is a polyphenol which the oral absorption in humans is about 75% and extensive metabolism in the intestine and liver results in an oral bioavailability considerably less than 1%. The results presented overcome such limitations, evidencing the mechanism of action and the advantages of developed nanoparticles as compared with resveratrol alone, both in vitro and in vivo.  The work is relevant, and the technology employed is adequate.

Global revision:

  • Figure legends present different formatting along the document, not fully complying with IJMS guidelines. Please revise them all.
  • English language should also be carefully revised along the entire document.
  • Safety of the different formulations in C6 cells is discussed, though safety to other healthy brain cells was not fully discussed. Is Pep-PP@Res specific for GBM and not toxic to other cells? Please include discussion, references and/or results in other brain cells to fully elucidate about this concern.
  • No discussion was included regarding the developed nanoparticles blood brain barrier permeability as comparatively with resveratrol alone. Should Pep-PP@Res be modified to include transferrin/transferrin antibody as well? Please discuss.

Minor errors:

Line 38: “Stupp Protocol [7], combination of tumor treatment fields (TTField) with temozolomide (TMZ), has been widely adopted in GBM therapy, because it increases 2-year 39 relative survival (RS) rates” – please consider revise the English language

Line 259: “By the end of treatment, 6 does of Res (..)” – please revise

Author Response

Please find the attached point-by-point response to your valuable comments, thanks.

Reviewer 2 Report

The authors state a significant reduction of in vivo tumor growth by reservatrol encapsulated  IL13R alpha  targeting nanoparticles. However, the STD is  quiet big and the number of animals  per treatment group is not indicated.

Reservatrol is an important modulator of the immune response, it can increase effector immune responses but also induce immunosuppression by for example inducing MDSC (in a dose dependent fashion). The authors should characterize the immuophenotype at least at the tumor site between Control and Treatment Groups, to demonstrate that tumor reduction is mediated  by an enhancement of the immune effector response and not due to a toxic response due to off target effects.

Author Response

Many thanks for your valuable comments. Attached please find the point-by-point response to the comments. 

Round 2

Reviewer 2 Report

Authors have not addressed the concerns of the author. The tumor growth curve is not significant . Immunophenotype of treated and untreated mice is not shown.

Round 3

Reviewer 2 Report

Please see comments to the Editor.